# Fortetropin supplementation prevents the rise in circulating myostatin but not disuse-induced muscle atrophy in young men with limb immobilization: A randomized controlled trial

**Changhyun Lim**, **James McKendry, Taylor Giacomin, Jonathan C. Mcleod, Sean Y. Ng, Brad S. Currier**, **Giulia Coletta**, **Stuart M. Phillips** *

Department of Kinesiology, Exercise Metabolism Research Group, McMaster University, Hamilton, Ontario, Canada

* phillis@mcmaster.ca

## Abstract

Supplementation with Fortetropin® (FOR), a naturally occurring component from fertilized egg yolks, reduces circulating myostatin concentration. We hypothesized that FOR would mitigate muscle atrophy during immobilization. We examined the effect of FOR supplementation on muscle size and strength during 2-wk of single-leg immobilization and recovery. Twenty-four healthy young men (22 ± 2 yrs; BMI = 24.3 ± 2.9 kg/m$^2$) were randomly allocated to either a Fortetropin® supplement (FOR-SUPP, n = 12) group consuming 19.8 g/d of FOR or placebo (PLA-SUPP, n = 12) group consuming energy- and macronutrient-matched cheese powder for 6-wk. The 6-wk period consisted of 2-wk run-in, 2-wk single-leg immobilization, and 2-wk recovery phase returning to habitual physical activities. Ultrasonography, dual-energy X-ray absorptiometry, muscle biopsies and isometric peak torque assessments were performed prior to and following each phase (days 1, 14, 28, and 42) to measure *vastus lateralis* and muscle fiber cross-section area (CSA), leg lean mass (LM), and muscular strength. Blood samples were taken on days 1 and 42 for measurement of plasma myostatin concentration, which increased in PLA-SUPP (4221 ± 541 pg/mL to 6721 ± 864 pg/mL, *P* = 0.013) but not in FOR-SUPP (5487 ± 489 pg/mL to 5383 ± 781 pg/mL, *P* = 0.900). After the immobilization phase, *vastus lateralis* CSA, LM, and isometric peak torque were decreased by 7.9 ± 1.7% (*P* < 0.001), -1.6 ± 0.6% (*P* = 0.037), and -18.7 ± 2.7% (*P* < 0.001) respectively, with no difference between groups. The decreased peak torque was recovered after 2-wk of normal activity (vs. day 1, *P* = 0.129); however, CSA and LM were not recovered (vs. day 1, *P* < 0.001 and *P* = 0.003, respectively), with no differences between groups. Supplementation with FOR prevented the rise in circulating myostatin but not disuse-induced muscle atrophy in young men after 2-wk of single-leg immobilization.

**Data Availability Statement:** All relevant data are within the manuscript and its Supporting information files.

**Funding:** This work was supported by funding from Myos Corporation (grant number: N/A, URL: http://myoscorp.com) to SMP. The funders had no role in study design, data collection and anlysis, decision to publish, or preparation of the manuscript.

**Competing interests:** I have read the journal's policy and the authors of this manuscript have the following competing interests: SMP reports grants or research contracts from the US National Dairy Council, Canadian Institutes for Health Research, Dairy Farmers of Canada, Roquette Freres, Ontario Centre of Innovation, Nestle Health Sciences, Myos, National Science and Engineering Research Council and the US NIH during the conduct of the study; personal fees from Nestle Health Sciences, non-financial support from Enhanced Recovery, outside the submitted work. SMP has patents licensed to Exerkine but reports no financial gains from any patent or related work. Other authors have declared that no competing interests exist. This does not alter our adherence to PLOS ONE policies on sharing data and materials.

## Introduction

Skeletal muscle is a plastic tissue that adapts its structure and metabolism in response to several internal and external variables, particularly mechanical load and protein ingestion [1]. Loss of muscle mass and function occurs rapidly with muscle disuse inherent to bed rest and single-leg immobilization [2, 3], which leads to impaired functional capacity, insulin resistance, and an increased risk for morbidity and mortality [4]. Nutritional support to counter disuse atrophy has included supplementation with essential amino acids [5] or omega-3 fatty acids [6], which have been shown to mitigate disuse-induced atrophy. Further work is warranted to identify effective strategies to maintain muscle mass and function during disuse, leveraging key molecular targets.

Myostatin (or growth differentiation factor-8, GDF-8) is a negative regulator of muscle growth and a member of the transforming growth factor-β family [7]. Various molecular mechanisms regulating protein synthesis and degradation are influenced by myostatin [8]. Myostatin also negatively regulates the activation and self-renewal of satellite cells (SC) by inhibiting the progress of SC into the S phase of the cell cycle [9]. In addition, myostatin reduces IGF-Akt pathway activity resulting in decreased protein synthesis initiation and increased translocation of Forkhead box protein O1 (Foxo1) into the nucleus. This mechanism releases E3 ubiquitin ligases, promoting muscle atrophy [8]. Previous studies reported an increased myostatin mRNA expression in skeletal muscle after 5 days of single-leg immobilization [10] and an increase in myostatin mRNA and protein expression after 3 days of unilateral lower limb suspension in healthy young men [11]. Whereas Jones et al. [12] showed no change in myostatin mRNA expression after 2-wk of single-leg immobilization, although there was an increased tendency of myostatin mRNA.

A previous study reported that 12-wk of Fortetropin® (FOR) ingestion, a non-thermal pasteurized, freeze-dried high protein egg yolk product, with resistance training lowered serum myostatin by 18–22% and increased muscle thickness by 3.7% and lean mass (LM) by 3.3% in trained young men, but placebo did not [13]. Moreover, consuming FOR reduced the expression of ubiquitin monomer protein and polyubiquitination genes and increased the activity of mechanistic target of rapamycin (mTOR) signaling after acute resistance exercise in a rodent model [13]. Evans et al. [14] also reported that muscle protein synthesis (MPS) was increased by 18% in older adults who consumed the FOR supplementation for 21 days compared with placebo; in the absence of changes in circulating myostatin. These previous studies support the notion that FOR may exert an anabolic influence on skeletal muscle. Interestingly, FOR ingestion may prevent disuse-induced muscle atrophy in dogs treated that consumed FOR during eight weeks of exercise restriction [15]. However, the impact of FOR supplementation on circulating myostatin and muscle tissue-level adaptations during muscle disuse in humans is unknown.

The primary aims of this trial were to determine whether FOR supplementation could prevent disuse-induced declines in muscle cross-sectional area (CSA) measured by ultrasonography and muscle biopsy (muscle fiber CSA), LM using dual-energy X-ray absorptiometry (DXA), and isometric strength during two weeks of single-leg immobilization phase. We hypothesized that FOR supplement would attenuate the decline in muscle size during immobilization, which would coincide with a reduction of circulating myostatin and alterations of molecular markers favoring a less catabolic state.

## Materials and methods

### Participants and ethics approval

A total of 24 young men (mean age of 22 ± 2 years old) with a body mass index (BMI) between 20 and 30 kg/m$^2$ were recruited for the study between January 2020 and February 2022 (This

trial was paused from March 2020 to August 2021 due to COVID-19-imposed restrictions on human research). Participants were non-smokers, generally healthy according to a medical screening questionnaire, and recreationally active but did not meet Canada's Physical Activity Guidelines (150 min of moderate-intensity exercise/week) and agreed not to consume eggs for the study duration. Exclusion criteria included severe orthopedic, cardiovascular, pulmonary, renal, liver, infectious disease, immune or metabolic disorder, consuming a vegan diet, taking medications known to affect protein metabolism (i.e., corticosteroids, non-steroidal anti-inflammatory drugs, or strength acne medication), and who had sustained a lower limb injury within a year prior to study participation. Before providing a written informed consent form, all potential participants were informed of the study purpose, experimental procedures, and possible risks. This study was approved by the Hamilton Integrated Research Ethics Board (project number: 7935) and conducted according to the principles expressed in the Declaration of Helsinki. This study was registered (http://clinicaltrials.gov/) as NCT05369026. A consolidated standards of reporting trials (CONSORT) flow diagram and chart can be seen in S1 and S2 Files, respectively.

Based on previous data [6, 16–18], the sample size for this protocol was calculated using G*power (Version 3.1.9.6, Germany) to achieve the expected declines in muscle cross-sectional area and assuming FOR would mitigate atrophy, an effect size of 0.33 (medium-sized effect, where 0.2 is low, 0.5 is medium, and 0.8 is high) [19], alpha was set at 0.05 and power at 0.9. The total sample size was 24 participants or 12/group (i.e., FOR vs. placebo).

## Experimental design

This study was a double-blind, randomized parallel group design (n = 12/group). Young men were randomly assigned into one of two groups, FOR-SUPP or PLA-SUPP group. FOR-SUPP group daily consumed 19.8 g of FOR for 6-wk, and PLA-SUPP consumed an energy- and macronutrient-matched cheese powder (placebo supplement). After the screening was performed, at least one week before beginning the study protocol, recruited participants were randomly assigned into the two groups based on a randomly generated sequence using http://www.randomization.com/. An investigator not directly involved with recruitment and testing held a key for the randomization. The study included a pre-treatment (run-in) phase (2-wk), an immobilization phase (2-wk), and a recovery phase (2-wk) for a total 6-wk. Participants were instructed not to alter their habitual physical activity habits and not to perform any intense physical activity, such as resistance exercise, for the duration of the study protocol. The 2-wk immobilization phase was accomplished using the knee brace model we have used previously [6, 16, 20]. Participants were worn a knee brace (X-ACT ROM knee brace, Don Joy, Dallas, TX, USA) on a randomly selected leg. Crutches were provided to allow for mobility throughout the immobilization phase, and the brace angle was adjusted so that participants' toes could not touch the ground during crutch-assisted ambulation. During the recovery phase, the knee brace was removed, and participants returned to their usual physical activity; however, participants were still required to refrain from any exercise. At 2-wk intervals from baseline (days 1, 14, 28, and 42), participants arrived at the laboratory at ~08:00 in the overnight fasted state and underwent procedures in the following order: DXA scan (following a urinary void) to assess body composition [20], ultrasonography for muscle CSA, blood sampling via venipuncture to assess circulating myostatin concentration and basic metabolic panel (blood urea nitrogen, BUN; creatine; glucose; alanine aminotransferase, ALT; aspartate aminotransferase, ASP; alkaline phosphatase, ALP; albumin; and bilirubin) to evaluate safety and tolerability of the compound (blood sampling were only performed at day 1 and 42), skeletal muscle biopsy to assess fiber type-dependent CSA, the number of myostatin-positive SC, and gene and protein

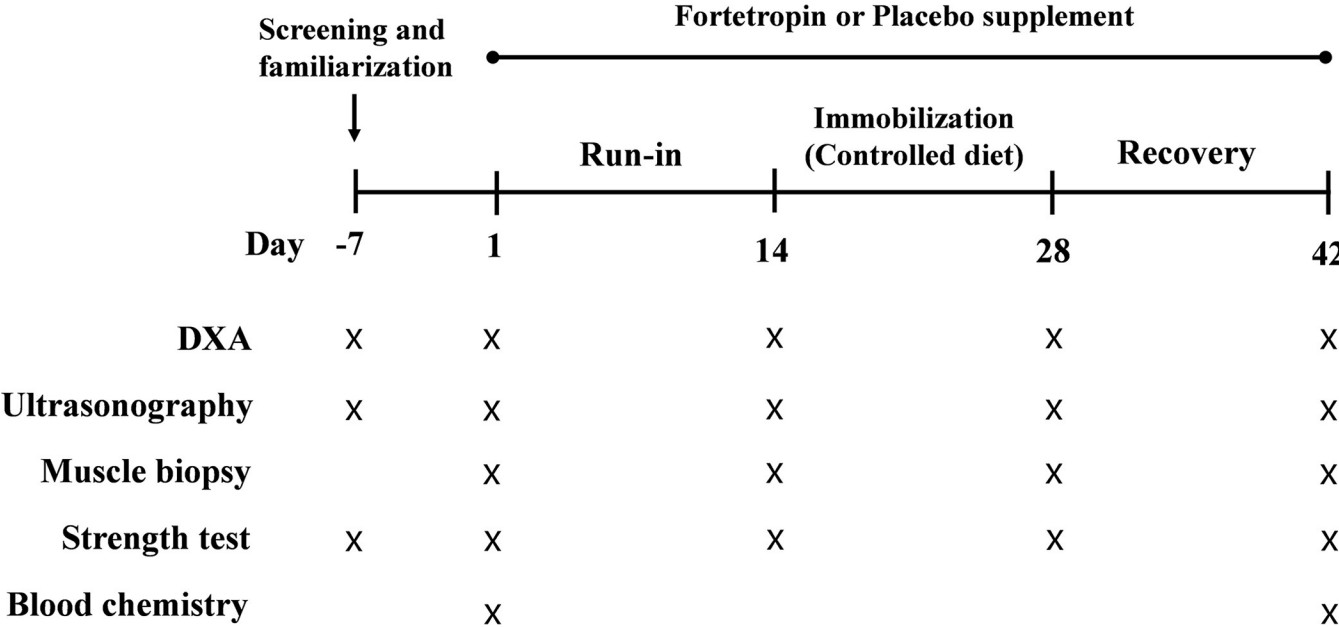

**Fig 1. Schematic overview of the study protocol.** Muscle biopsies, strength tests, and ultrasonography were performed on the immobilized leg only; DXA, dual-energy X-ray absorptiometry.

expression related to protein synthesis and degradation, and isometric knee-extensor torque test to assess leg strength. Specific details of the study protocol are presented in Fig 1.

## Nutritional supplements

FOR is a non-thermal pasteurized, freeze-dried high proteo-lipid, fertilized egg yolk product. It comprises 2.4% carbohydrates, 59% fat, 34% protein, and 3.5% total ash, with an energy content of 6.7 kcal/g. The ingredients of FOR and placebo were analyzed using Association of Official Agricultural Chemists (AOAC) method from a food analysis laboratory (Certified Laboratories Inc., NY, USA). Participants in FOR-SUPP consumed 19.8 g of FOR daily with breakfast, and the PLA-SUPP group consumed 19.8 g of cheese powder that was matched for macronutrient and energy to FOR-SUPP. The supplements were provided pre-mixed into a chocolate pudding to match the flavor and texture to maintain blinding.

## Diet and physical activity

Participants were required not to consume alcohol, eggs, or other supplements throughout the protocol to rule out any possibility that could interfere with the FOR effects. Besides that, participants were instructed to consume their regular diets during run-in (days 1 to 14) and the recovery phase (days 28 to 42) and complete a 3-day dietary log (2 weekdays and 1 weekend day) to evaluate their total daily energy, carbohydrate, protein, and fat intake. These values were analyzed using NutriBase software (Cybersoft Inc., version 11.5, Pheonix, AZ, USA). We provided standardized meals that consisted of 1.2 g of protein per body mass per day throughout the two-week immobilization phase. The daily energy requirements of the participants were determined using the Harris-Benedict equation, with appropriate adjustments for physical activity levels (1.5, light activity). Thus, all participants consistently consumed sufficient dietary protein for muscle protein synthesis [21, 22] and tried to minimize diet-induced variability. Standardized meals were provided as pre-packaged frozen meals (Heart-to-Home

Meals, Hamilton, ON, Canada), and the meal plans were customized according to each participant's personal food preferences.

Participants were required to refrain from intense physical activity, including resistance exercise, and not alter their habitual physical activity throughout the protocol. Participants wore an Actigraph wGT2X-BT activity monitor (ActiGraph, Pensacola, FL, USA) on the wrist of their dominant arm to assess their physical activity level (steps/d; physical activity, kcal/d; metabolic equivalents of task [METs/d]) during the study.

## Ultrasonography

The CSA of *vastus lateralis* measured by B-module ultrasonography was validated as a suitable alternative to MRI, which is considered the gold standard for measuring muscle size, for measuring changes in muscle size in response to disuse atrophy [23]. Using this method, with a 18L5 probe (BK Medical North America, Peabody, MA, USA), we measured the CSA of *vastus lateralis* of the leg randomly assigned for immobilization at days 1, 14, 28, and 42. The procedure involved participants laying supine for more than 10 min to normalize fluid shifts in the body, and we marked the location for imaging, 50% of the distance between the top of the patellar bone and the greater trochanter of the femur, identified by palpation. Afterward, a straight line was drawn along the longitudinal of the *vastus lateralis*, and the images were taken at 2 cm increments along the line resulting in ~7–9 images. The same investigator collected images during each visit to avoid variability in pressure amongst the images. Acquired images were stitched together using Gnu Image Manipulation Program (GIMP, version 2.9.22, Mountain View, CA, USA) to create a successive image of the *vastus lateralis* by aligning subcutaneous fat, superficial and deep aponeuroses, and intramuscular fat deposits between images (S1 Fig). These images were blinded to both group and time, and the CSA of *vastus lateralis* was analyzed using the Polygon tracing tool in ImageJ.

## Knee extensor isometric muscle torque

After a familiarization session on the screening day, unilateral isometric knee-extensor torque was measured using the BIODEX (System 3, Biodex Medical Systems, Shirley, NY, USA) on days 1, 14, 28, and 42. The dynamometer was set to 60˚ from the resting (90˚) position, and participants performed four maximal voluntary isometric leg extensions, lasting 5 seconds for 4 repetitions, 2 min apart. The average of the highest recorded 3 trials was used for maximal strength per visit.

## Blood measurements

Blood samples were taken from an antecubital vein on days 1 and 42 and collected in EDTA tubes to aliquot plasma. The collected blood samples were stored on ice for 30 min before centrifuging at 1000 g for 10 min at 4˚C, and plasma was aliquoted into Eppendorf tubes for analysis. A commercial GDF-8/Myostatin enzyme-linked immunosorbent assay (ELISA) kit (R&D systems, Minneapolis, NM, USA) was used to assess circulating myostatin concentration. Liver enzymes, ALT, ALP, and AST were measured using UV absorbance (Roche Cobas 8000 System, Indianapolis, IN, USA), and BUN, creatinine, glucose, albumin, and bilirubin were measured by UV-Visible Spectrophotometer-CHEM7 by the McMaster University core clinical laboratory services. All assays had an intra-assay CV of less than 3.5%

## Muscle biopsy

During the study, four muscle biopsies were collected (days 1, 14, 28, and 42) from the *vastus lateralis* of the leg that was assigned for immobilization. The first biopsy was collected from approximately 15 cm above the patella, and the subsequent biopsies were collected from ~2 or 3 cm apart between them to avoid the effect of pre-sampling. All muscle biopsies were obtained using a 5-mm Bergstrom needle, custom modified for manual suction under 1% xylocaine local anesthesia. Through manual dissection, muscle tissues were carefully freed from visible connective tissue, fat, and blood. A piece (~40 mg) of the muscle tissue was embedded in optimal cutting temperature compound (OCT, Tissue-Tek, The Netherlands) for immunohistochemical analysis. The rest of the muscle tissue was snap-frozen in liquid nitrogen for gene and protein expression analysis. All muscle samples was stored at -80˚C until analysis.

## Immunohistochemistry

Muscle samples embedded in the OCT compound were cut into 5 μm sections in a cryostat at -20˚C and transferred to the Super Frost Plus microscope slide (Fisher brand, Pittsburgh, PA, USA). Tissue sections were fixed in 4% paraformaldehyde solution for 5 min, permeabilized in 100% methanol for 10 min at– 20˚C and 1% triton/phosphate-buffered saline (TPBS) for 10 min. Tissue sections were blocked for 30 min at room temperature in 10% goat serum in TPBS. To assess fiber type dependent CSA, the blocked tissue sections were incubated with primary antibodies against myosin heavy chain (MHC) I (BA.D5, 1:50, DSHB, Iowa City, IA, USA), MHC II (SC-71, 1:50, DSHB, Iowa City, IA, USA), and laminin (ab11575, 1:100, Abcam, Cambridge, MA, USA), and for analysis of the number of SC and Myostatin-positive SC (MSTN+/PAX7+), Pax7 (MAB1675, 1:100, R&D Systems, Minneapolis, MN, USA), Myostatin (AB3239, 1:100, Millipore, Etobicoke, ON, Canada), MHC II (ab91506, 1:1000, Abcam, Cambridge, MA, USA), and laminin (ab11575, 1:100, Abcam, Cambridge, MA, USA) were applied for overnight at 4˚C. Appropriate secondary antibodies were applied; for CSA analysis, Alexa 647 goat anti-mouse IgG2b (A21242, 1:250, Invitrogen, Waltham, MA, USA), Alexa 488 goat anti-mouse IgG1 (A21121, 1:500, Invitrogen, Waltham, MA, USA), and Alexa 350 goat anti-rabbit IgG (H+L) (A11046, 1:500, Invitrogen, Waltham, MA, USA), and for SC and MSTN+/PAX7+, Alexa 594 (A11005, 1:500, Invitrogen, Waltham, MA, USA), Alexa 647 (A21244, 1:500, Invitrogen, Waltham, MA, USA), and Alexa 488 (A11008, 1:500, Invitrogen, Waltham, MA, USA). Nuclei were stained with 4,6-diamidino-2-phenylindole (1:2000, Sigma-Aldrich, Oakville, ON, Canada). Stained samples were imaged by a Nikon Eclipse Ti2pad microscope (Nikon Instruments, Melville, NY, USA) with a high-resolution Photometrics Moment 7 Megapixel CMOS camera (Teledyne Photometrics, Tucson, AZ, USA) with 20 times objective and analyzed using Nikon NIS element AR software (Nikon Instruments). All analyses were conducted on more than 100 fibers per fiber type per participant per time point [24, 25], and assessors were blinded to both group and time.

## Western blotting

Phosphorylation/total protein content of mTOR and downstream proteins (p70S6K, S6K, and 4EBP1) were analyzed at pre- (day 14) and post-immobilization (day 28) to investigate the anabolic effect of FOR supplement during immobilization. After snap frozen muscle tissues were homogenized with ice-cold lysis buffer (10 μL·mg$^{-1}$; 25 mM Tris 0.5% vol/ vol Triton X-100 and protease/phosphatase inhibitor cocktail tablets; Complete Protease inhibitor Mini-Tabs [Roche] and aPhosSTOP [Roche Applied Science]), each sample was centrifuged at 11,000 g for 10 min at 4˚C. The protein concentration of the supernatants was mixed with 4 × Laemmli

buffer containing 10% 2-mercaptoethanol and distilled water to 2 μg/μL of protein concentration. 10 μg of each sample was loaded into wells on 4%–15% TGX Stain-Free Precast Gels (Bio-Rad, Hercules, CA, USA) with a protein ladder (Precision Plus Protein Standard, Bio-Rad, Hercules, CA, USA) and an internal standard calibration curve. Gel electrophoresis was run at 200 V for 45 min at room temperature. After UV activation of the gels, the proteins were transferred to a nitrocellulose membrane by turbo transfer. UV light activation revealed the total protein content on the nitrocellulose membrane, which was evenly loaded among each lane. Membranes were blocked with 5% bovine serum albumin for 90 min at room temperature. Primary antibodies were applied for overnight at 4°C; 1:1000, Cell Signaling Technology, Danvers, MA, USA; total form: mTOR (2972S), p70S6K (9202L), S6K (2217L), and 4EBP1 (9644S); phosphorylation form: $mTOR^{Ser2448}$ (5536S), $p70S6K^{Thr389}$ (9205L), $S6K^{ser235/236}$ (2211S), and $4EBP1^{Thr37/46}$ (2855C). Following Tris-buffered saline and Tween 20 (TBS-T, Millipore, Oakville, ON, Canada) washes, membranes were incubated in appropriate secondary antibodies (anti-rabbit IgG, 7074S, Cell Signaling Technology, Dancers, MA, USA) for 90 min at room temperature. Proteins of interest were detected by chemiluminescence solution (Clarity Western ECL substrate, Bio-Rad, Hercules, CA, USA) and the ChemiDoc MP Imaging system and analyzed by Image Lab Software 6.0.1. Western blotting was performed only on a subgroup of participants (9 of FOR-SUPP and 6 of PLA-SUPP) due to shortage of muscle tissue.

## RNA isolation, reverse transcription, and real-time quantitative PCR

E3 ubiquitin ligases [muscle specific F-box protein (Atrogin-1) and muscle RING-finger protein 1 (MuRF1)] were analyzed pre- (day 14) and post-immobilization (day 28) to investigate protein degradation through the ubiquitin-proteasome system (UPS) during immobilization. Total RNA was isolated from 20 mg of snap-frozen muscle tissues using the TRIzol reagent (15596–026, Invitrogen, Waltham, MA, USA) according to the manufacturer's protocol. RNA was resuspended in 33 μL of RNase-free distilled water (UltraPure water, Invitrogen, Waltham, MA, USA), and RNA concentration and purity were measured using Nano-Drop 2000 Spectrophotometer (Thermo Fisher Scientific Rockville, MD, USA). cDNA was synthesized by reverse transcription using a High-Capacity cDNA Reverse Transcription Kit (Applied Biosystems, Foster City, CA, USA). 20 ng of the synthesized cDNA samples were run with triplicate 6 μL reactions containing PCR master mix (A6002, GoTaq, Promega, Madison, WI, USA). 5uM of primers (Sigma-Aldrich) of target genes [sequence 5' to 3'; Atrogin-1, forward: `GCAGCTGAACAACATTCAGATCAC`, reverse: `CAGCCTCTGCATG−ATGTTCAGT`; MuRF1, forward: `CCTGAGAGCCATTGACTTTGG`, reverse: `CTTCCCTTC−TGTGGACTCTTCCT`] using QuantStudio 5 Real-Time PCR system (Applied Biosystems, Foster City, CA, USA). Data were analyzed using the comparative $2^{-\Delta\Delta Ct}$ method [26]. 18S (forward: `TTCGGAACTGAGGCCGCC ATGAT`, reverse: `TTTCGCTCTGGTCGCTCTTG`) was employed as the housekeeping gene since no differences between groups (FOR and PLA) and within the time points (pre and post) were detected.

## Statistical analysis

All statistical analyses were performed using SPSS Statistics Software for Windows, version 26.0 (IBM Corp., Armonk, NY, USA). The Shapiro-Wilk test was performed to assess normality for the distribution of data. Non-normally distributed data (the number of SC in type I and II and the phosphorylation ratio to total mTOR and S6K) were log-transformed [27] and showed normal data distribution. Independent t-tests were performed to compare participants' baseline characteristics and the percentage change in plasma myostatin concentration

**Table 1. Baseline characteristics of the participants.**

|  | FOR-SUPP (n = 11) | PLA-SUPP (n = 9) | P |
|---|---|---|---|
| Age (years) | 22 ± 2 | 22 ± 1 | 0.74 |
| Height (cm) | 178 ± 5 | 175 ± 8 | 0.33 |
| Body mass (kg) | 75.4 ± 6.3 | 77.9 ± 15.8 | 0.05 |
| BMI (kg/m$^2$) | 23.6 ± 2.0 | 25.4 ± 3.4 | 0.09 |
| Total body LM (kg) | 57.6 ± 5.6 | 55.5 ± 9.3 | 0.48 |
| Immobilized leg LM (kg) | 11.0 ± 1.1 | 10.5 ± 1.7 | 0.48 |

Data are expressed as mean ± standard deviation. FOR-SUPP, Fortetropin® supplement; PLA-SUPP, placebo supplement; BMI, body mass index; LM, lean mass.

between groups. Two-way repeated-measures analysis of variance (ANOVA) was used to assess the group (FOR-SUPP vs. PLA-SUPP) by time (day 1, 14, 28, and 42; run-in, immobilization, and recovery phase), and a Tukey's post hoc test was performed for following analysis of the significant interactions from the ANOVA analysis. Pearson's correlation coefficients were performed to assess the association between the change in myostatin concentration and CSA measured by ultrasonography. Statistical significance was $P < 0.05$, and all data are presented as mean ± standard deviation (SD).

## Results

### Participant's baseline characteristics

In total, 24 participants were recruited, and 22 completed the protocol; 2 participants dropped out due to the Covid-19 outbreak. Of the participants who completed the study, 2 participants in the PLA-SUPP were excluded due to poor compliance with wearing the knee brace during the immobilization phase (S1 File). The baseline characteristics of the participants are presented in Table 1.

### Physical activity

Physical activity variables are shown in Table 2. During the immobilization phase, steps, energy expenditure by physical activity, and METs were significantly decreased compared to the run-in phase (step, $P < 0.001$; energy expenditure by physical activity, $P < 0.001$; METs, $P < 0.001$) with no difference between groups (step, $P = 0.989$; energy expenditure by physical activity, $P = 0.921$; METs, $P = 0.304$), and the decreased level of the physical activity variables were fully recovered to the level of the run-in phase during the recovery phase (step, $P < 0.001$; energy expenditure by physical activity, $P < 0.001$; METs, $P < 0.001$).

**Table 2. Physical activity during each phase.**

|  | Run-in | | Immobilization | | Recovery | |
|---|---|---|---|---|---|---|
|  | FOR-SUPP | PLA-SUPP | FOR-SUPP | PLA-SUPP | FOR-SUPP | PLA-SUPP |
| Steps/d | 7782 ± 2488 | 7992 ± 3870 | 5550 ± 2232* | 4591 ± 3156* | 7092 ± 2296 | 7786 ± 3883 |
| Energy expenditure by physical activity (kcal/d) | 933 ± 356 | 972 ± 545 | 742 ± 340* | 698 ± 548* | 852 ± 320 | 924 ± 627 |
| METs/d | 1.5 ± 0.2 | 1.6 ± 0.3 | 1.3 ± 0.1* | 1.3 ±0.2* | 1.4 ± 0.1 | 1.6 ± 0.3 |

Data are expressed as mean ± standard deviation. FOR-SUPP, Fortetropin® supplement; PLA-SUPP, placebo supplement; * $P < 0.05$, significantly different from run-in and recovery phase (main effect for time).

### *Vastus lateralis* CSA and LM of the immobilized leg

The results of *vastus lateralis* CSA and LM of the immobilized leg are presented in Fig 2, and the representative image of *vastus lateralis* cross-section obtained by ultrasonography is shown in S1 Fig. There were no interactions between group and time for the absolute value of CSA ($P = 0.763$) and the percentage change in CSA ($P = 0.487$), but main effect for time in absolute value of CSA ($P < 0.001$) and the percentage change in CSA ($P < 0.001$). In the analysis of main effect for time, the mean of absolute CSA from both groups was $28.6 \pm 1.3$ cm$^2$ and $28.6 \pm 1.2$ cm$^2$ on day 1 and 14, respectively, and declined to $26.1 \pm 1.1$ cm$^2$ at day 28 (Day 14 vs. 28, $P < 0.001$) with no difference between groups ($P = 0.351$), and the decreased CSA did not return after recovery phase (day 1 vs. 42, $P < 0.001$; Fig 2A). The percentage change in CSA from day 1 to the end of immobilization (day 28) and the recovery phase (day 42) were– $7.9 \pm 1.7\%$ (vs. run-in, $P < 0.001$) and– $8.0 \pm 1.8\%$ (vs. run-in, $P < 0.001$), respectively

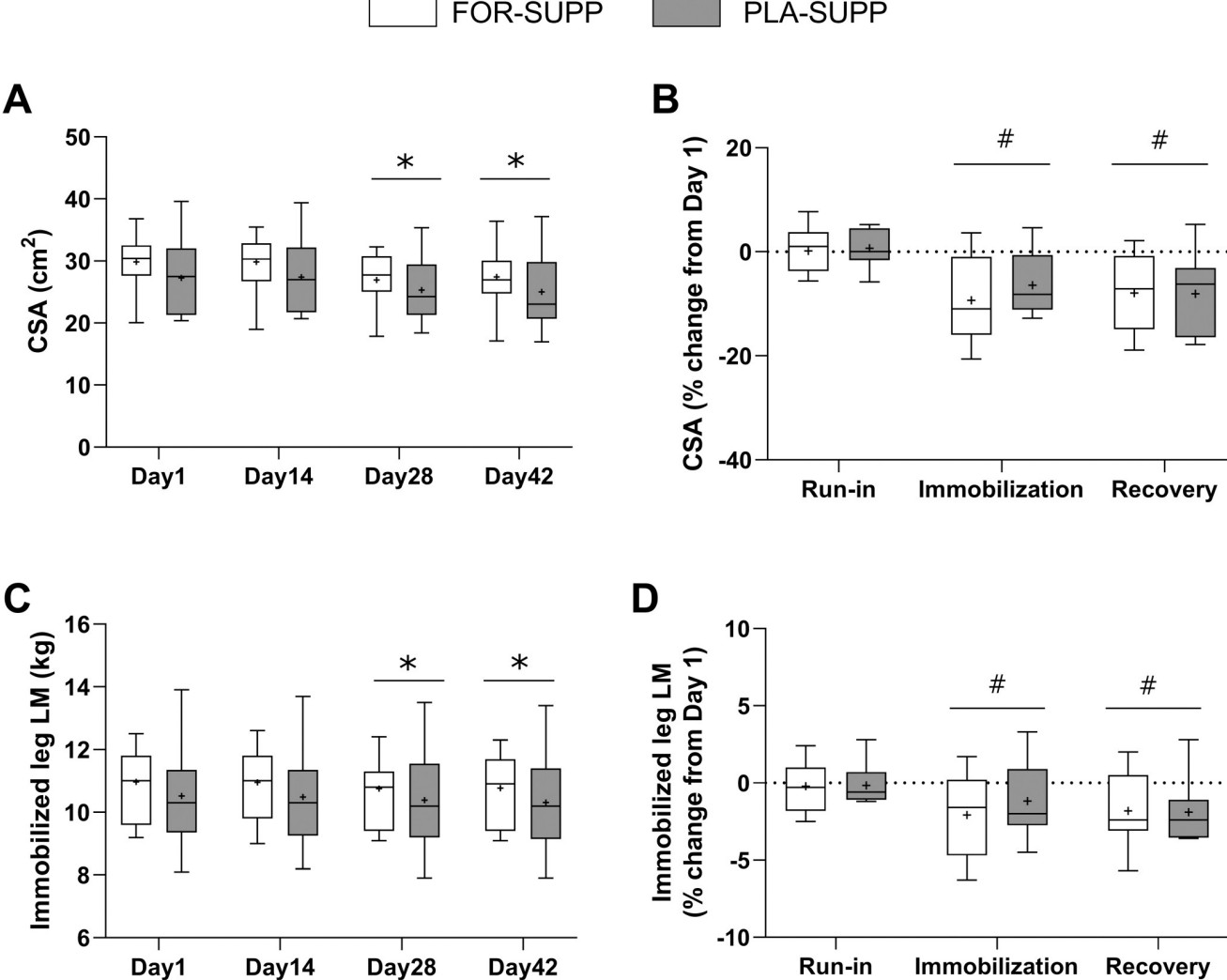

**Fig 2.** The absolute value of *vastus lateralis* CSA (A) and lower limb LM (C) of the immobilized leg at each phase, and the percentage change in *vastus lateralis* CSA (B) and lower limb LM (D) during each phase. Boxes represent the 25$^{th}$ to 75$^{th}$ percentile, with the central horizontal lines indicating the median. The whiskers represent the maximum and minimum values, and the cross represents the mean value. FOR-SUPP, Fortetropin® supplement; PLA-SUPP, placebo supplement; CSA, cross-sectional area; LM, lean mass. * $P < 0.05$, significant difference from Day 1 and 14 (main effect), # $P < 0.05$, significant difference from run-in (main effect).

(Fig 2B). There was no interaction between the group and time for the immobilization leg LM ($P = 0.870$) and the percentage change in the LM ($P = 0.740$), but main effect for time in the immobilization leg LM ($P = 0.015$) and the percentage change in LM ($P = 0.041$). In the analysis of main effect for time, the mean of immobilization leg LM from both groups decreased from 10.76 ± 1.37 kg and 10.74 ± 1.33 kg at day 1 and 14, respectively, to 10.58 ± 1.37 kg at day 28 (Day 1 vs. 28, $P = 0.015$; Day 14 vs. 28, $P = 0.043$) with no difference between groups ($P = 0.499$), and the decreased leg LM was not recovered after recovery phase (day 1 vs. 42, $P = 0.003$; Fig 2C). The percentage changes in the immobilized leg LM from day 1 to the end of the immobilization and recovery phase were -1.6 ± 0.6% (vs. run-in, $P = 0.037$) and -1.8 ± 0.5% (vs. run-in, $P = 0.018$), respectively (Fig 2D).

## Muscle fiber CSA

The representative images of fiber type-dependent CSA are shown in S2 Fig. There were no interactions between group and time for absolute CSA ($P = 0.659$), and the percentage change in CSA ($P = 0.646$) of Type I fibers (Fig 3A and 3B), but main effect for time in the percentage change in Type I fibers CSA ($P = 0.033$). In the analysis of main effect for time, Type I CSA decreased by 6.5 ± 3.8% after the immobilization phase (vs. run-in, $P = 0.036$) with no difference between groups ($P = 0.669$) and did not recover after the recovery phase (vs. run-in, $P = 0.853$; Fig 3B). There were no interactions between group and time for absolute CSA ($P = 0.940$) and the percentage change in CSA ($P = 0.680$) of type II fibers (Fig 3C and 3D), but main effect for time in the absolute ($P = 0.033$) and the percentage change ($P = 0.024$) CSA in Type II fibers. In the analysis of main effect for time, Type II CSA decreased from 5113 ± 1185 $\mu m^2$ and 5091 ± 1205 $\mu m^2$ at day 1 and 14, respectively, to 4555 ± 1038 $\mu m^2$ at day 28 (Day 1 vs. 28, $P = 0.027$; Day 14 vs. 28, $P = 0.010$) with no difference between groups ($P = 0.544$) and recovered to 4806 ± 1206 $\mu m^2$ at day 42 (day 1 vs. 42, $P = 0.182$). The percentage changes in Type II CSA from day 1 to the end of the immobilization and recovery phase were -8.8 ± 4.1% (vs. run-in, $P = 0.023$) and -5.0 ± 3.8% (vs. run-in, $P = 0.080$), respectively, with no difference between groups ($P = 0.567$).

## Isometric knee-extensor torque

There were no group-by-time interactions for the absolute value of isomeric knee-extensor peak torque ($P = 0.579$) or the percentage change in the peak torque ($P = 0.379$; Fig 4), but main effect for time in peak torque ($P < 0.001$) and the percentage change in the peak torque ($P = 0.003$). In the analysis of main effect for time, the absolute value of peak torque decreased from 160 ± 10 Nm and 162 ± 11 Nm at days 1 and 14 to 129 ± 9 Nm at day 28 (day 1 vs. 28, $P < 0.001$; day 14 vs. 28, $P < 0.001$) with no difference between groups ($P = 0.638$) and recovered to 152 ± 9 Nm at day 42 (day 1 vs. 42, $P = 0.129$) (Fig 4A). The percentage change in peak torque decreased from day 1 to the end of the immobilization and recovery phases were -18.7 ± 2.7% (vs. run-in, $P < 0.001$) and -3.5 ± 3.7% (vs. run-in, $P = 0.196$) with no difference between groups ($P = 0.327$; Fig 4B).

## Blood variables

The values for the blood variables are presented in Table 3 and Fig 5. There was a significant difference in ALP ($P = 0.027$) by ANOVA; however, *post hoc* analysis did not reveal significant pairwise differences between or within groups at any time point ($P > 0.05$; Table 3). There were no group-by-time interactions for BUN ($P = 0.977$), creatinine ($P = 0.991$), glucose ($P = 0.267$), ALT ($P = 0.294$), ASP ($P = 0.275$), albumin ($P = 0.329$), and bilirubin ($P = 0.429$)

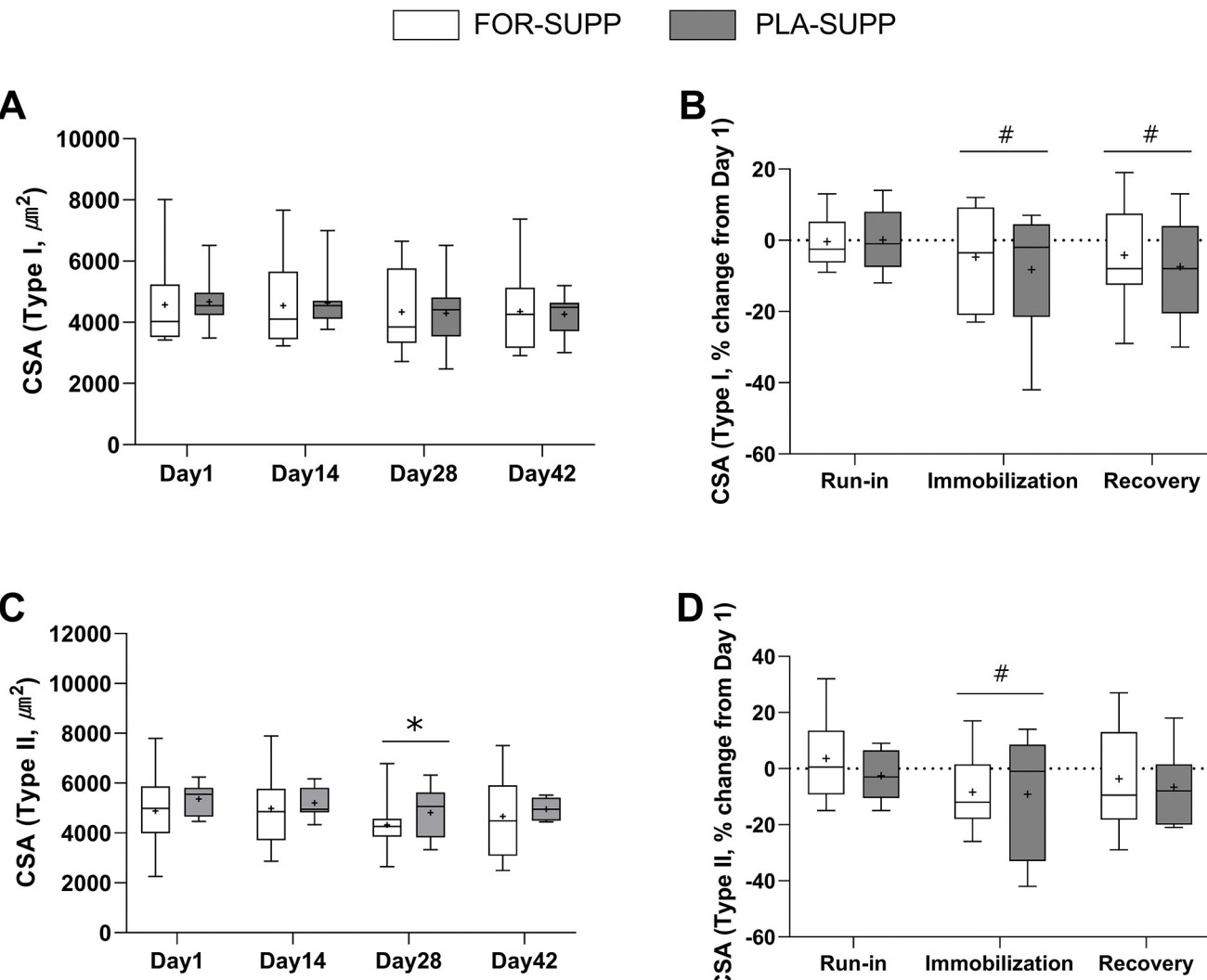

**Fig 3.** CSA of Type I (A) and Type II fiber (C) of the immobilized leg at each phase, and the percentage change in CSA of Type I (B) and Type II fiber (D) during each phase. Boxes represent the 25th to 75th percentile, with the central horizontal lines indicating the median. The whiskers represent the maximal and minimal values, and the cross represents the mean value. FOR-SUPP, Fortetropin® supplement; PLA-SUPP, placebo supplement; CSA, cross-sectional area. * $P < 0.05$, significantly different from Day 1 and 14 (main effect), # $P < 0.05$, significantly different from run-in (main effect).

(Table 3). ASP decreased from 24.9 ± 1.2 U/L on day 1 to 20.7 ± 1.7 U/L on day 42 ($P = 0.047$) with no difference between groups ($P = 0.289$) (Table 3).

There was no difference in myostatin concentration in plasma between day 1 and 42 in FOR-SUPP (5487 ± 489 pg/mL to 5382 ± 781 pg/mL, $P = 0.900$); however, in PLA-SUPP, myostatin concentration in plasma increased from 4221 ± 541 pg/mL at day 1 to 6721 ± 86 pg/mL at day 42 ($P = 0.013$) (Fig 5A). The percentage change in plasma myostatin concentration between days 1 and 42 was 4 ± 32% in FOR-SUPP and 75 ± 89% in PLA-SUPP (FOR-SUPP vs. PLA-SUPP, $P = 0.005$) (Fig 5B). There was a statistically significant correlation between the percentage change in CSA measured by ultrasonography from day 1 to 42 and post-plasma myostatin concentration with medium association (r = 0.469, $P = 0.037$) (Fig 5C). Despite no statistically significant correlation ($P = 0.066$), the correlation between the percentage change in CSA and myostatin concentration from day 1 to 42 showed a medium association (r = 0.419) (Fig 5D).

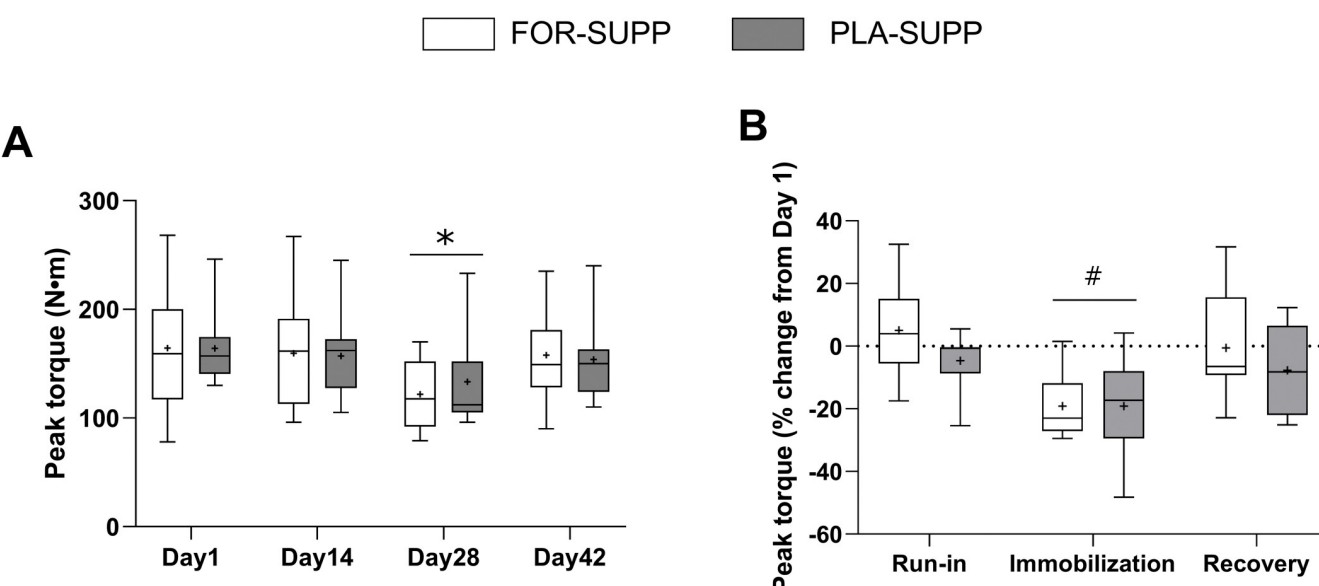

**Fig 4.** Peak torque (A) of the immobilized leg at each phase and the percentage change in peak torque (B) during each phase. Boxes represent the 25th to 75th percentile, with the central horizontal lines indicating the median. The whiskers represent the maximal and minimal values, and the cross represents the mean value. FOR-SUPP, Fortetropin® supplement; PLA-SUPP, placebo supplement. * $P < 0.05$, significantly different from Day 1 and 14 (main effect for time), # $P < 0.05$, significantly different from run-in and recovery (main effect for time).

## SC and myostatin-positive SC content

The results of SC (PAX7+) and myostatin-positive SC (MSTN+/PAX7+) are presented in Fig 6. There were no interactions between group and time for the number of SC in type I ($P = 0.976$, Fig 6B) and II fiber ($P = 0.894$, Fig 6C) and the percentage of myostatin-positive SC per SC in type I ($P = 0.433$, Fig 6D) and II fiber ($P = 0.938$, Fig 6E). With no difference between groups ($P = 0.813$), the percentage of myostatin-positive SC in type II fiber increased from 35 ± 5% and 42 ± 5% on day 1 and 14, respectively, to 55 ± 5% at day 28 (Day 1 vs. 28, $P = 0.001$; Day 14 vs. 28, $P = 0.022$), and the increased the percentage of myostatin-positive SC was back to normal level (36 ± 4%) at day 42 (Day 1 vs. 42, $P = 0.903$) (Fig 6E).

**Table 3. Blood variables at day 1 and 42.**

|  | FOR-SUPP | | PLA-SUPP | |
|---|---|---|---|---|
|  | Day 1 | Day 42 | Day 1 | Day 42 |
| BUN (mM) | 5.2 ± 0.9 | 5.1 ± 1.0 | 5.3 ± 0.8 | 5.3 ± 0.5 |
| Creatinine (μM) | 91.0 ± 13.1 | 90.8 ± 9.2 | 83.9 ± 18.9 | 83.5 ± 12.7 |
| Glucose (mM) | 4.5 ± 0.5 | 4.8 ± 0.4 | 5.1 ± 0.4 | 5.0 ± 0.4 |
| ALT (U/L) | 16.0 ± 6.8 | 19.0 ± 9.0 | 22.2 ± 11.1 | 18.2 ± 13.3 |
| ASP (U/L) | 22.6 ± 3.8 | 20.6 ± 9.1* | 27.2 ± 7.1 | 20.8 ± 4.3* |
| ALP (U/L) | 57.4 ± 13.2 | 63.3 ± 15.6 | 66.4 ± 18.7 | 56.6 ± 14.7 |
| Albumin (g/L) | 42.4 ± 4.9 | 41.4 ± 4.3 | 41.7 ± 3.4 | 42.4 ± 5.2 |
| Bilirubin (μM) | 19.5 ± 5.1 | 17.8 ± 8.4 | 18.6 ± 6.8 | 18.9 ± 6.9 |

Data are expressed as mean ± standard deviation. FOR-SUPP, Fortetropin® supplement; PLA-SUPP, placebo supplement; BUN, blood urea nitrogen; ALT; alanine aminotransferase; ASP, aspartate aminotransferase; ALP, alkaline phosphatase; * $P < 0.05$, significant difference from Day 1 (Main effect).

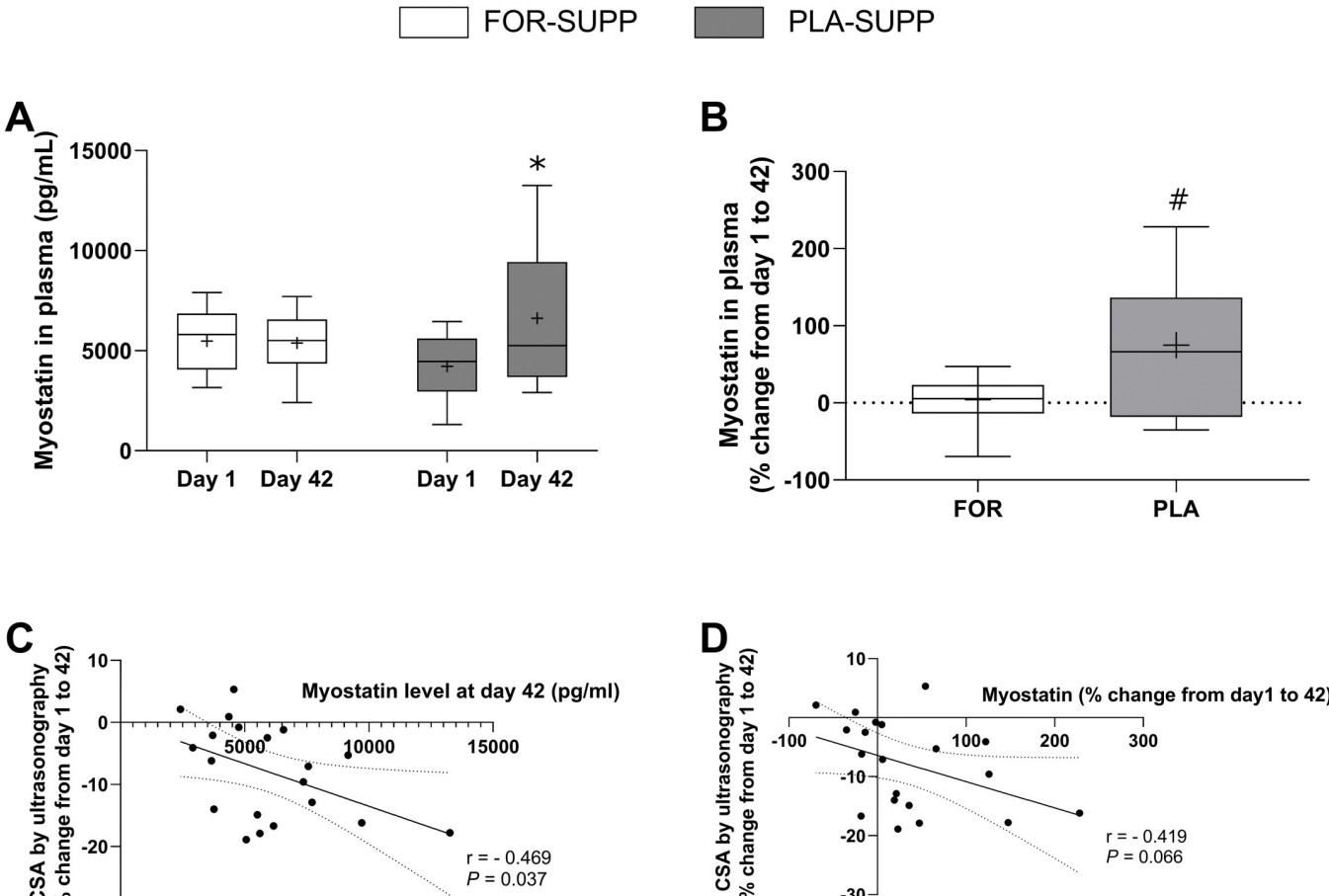

**Fig 5.** Plasma myostatin concentration at days 1 and 42 (A) and the percentage change in plasma myostatin concentration from day 1 to day 42 (B). The scatterplots show the correlation between the percentage change in CSA by ultrasonography and myostatin circulation concentration at day 42 (C), and the percentage change in plasma myostatin concentration (D). Boxes represent the $25^{th}$ to $75^{th}$ percentile, with the central horizontal lines indicating the median. The whiskers represent the maximal and minimal values, and the cross represents the mean value. FOR-SUPP, Fortetropin® supplement; PLA-SUPP, placebo supplement; CSA, cross-sectional area. * $P < 0.05$, significantly different from pre within the group, # $P < 0.05$, significantly different from FOR.

## Gene and protein expression

There were no interactions between group and time for E3 ubiquitin ligases, Atrogin-1 ($P = 0.776$, Fig 7A) and MuRF1 ($P = 0.962$, Fig 7B). There were no interactions between group and time for the ratio of phosphorylated to total (P/T) mTOR ($P = 0.522$, Fig 7C), S6K ($P = 0.640$, Fig 7E), and 4EBP1 ($P = 0.160$, Fig 7F). However, P/T p70S6K at post time point increased by 38 ± 46% compared to pre in FOR only ($P = 0.032$, Fig 7D).

## Discussion

In the present study, the effect of FOR supplementation were compared with a macronutrient- and energy-matched placebo on indices of muscle size and strength during two weeks of single-leg immobilization in young, healthy men. We discovered that ingestion of FOR prevented the rise in circulating plasma myostatin. However, FOR supplementation did not attenuate the loss of muscle CSA measured by ultrasonography, muscle fiber CSA in type I and II, and LM of the immobilized leg. In addition, both groups did not recover the immobilization-induced reduction in muscle CSA and LM of the immobilized leg after two weeks of returning to

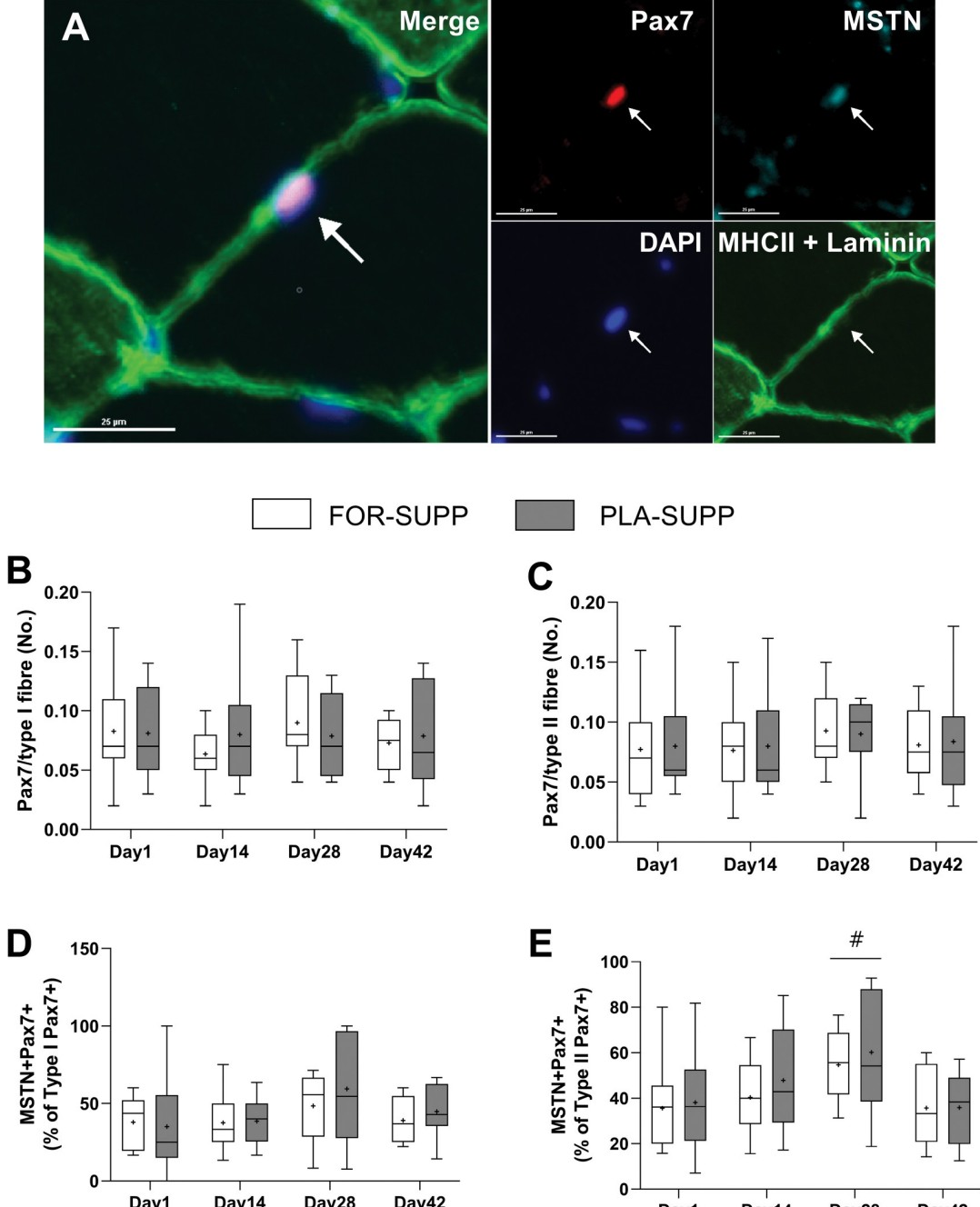

**Fig 6.** The representative image of Pax7 (red), MSTN (teal), DAPI (blue), MHCII and laminin (green), and merge of all the markers (A). The number of SC (PAX7+) in type I fiber (B) and type II fiber (C), and the percentage of myostatin-positive SC (MSTN+/Pax7+)/SC (Pax7+) in type I fiber (D) and type II fiber (E). The scale bar is 25 μm, and a white arrow indicates a myostatin-positive SC (MSTN+/Pax7+). Boxes represent the 25th to 75th percentile, with the central horizontal lines indicating the median. The whiskers represent the maximal and minimal values, and the cross represents the mean value. FOR-SUPP, Fortetropin® supplement; PLA-SUPP, placebo supplement; MSTN, myostatin; SC, satellite cells; DAPI, 4,6-diamidino-2-phenylindole; MHCII, myosin heavy chain II; # $P < 0.05$, significant difference from Day 1, 14 and 42 (main effect for time).

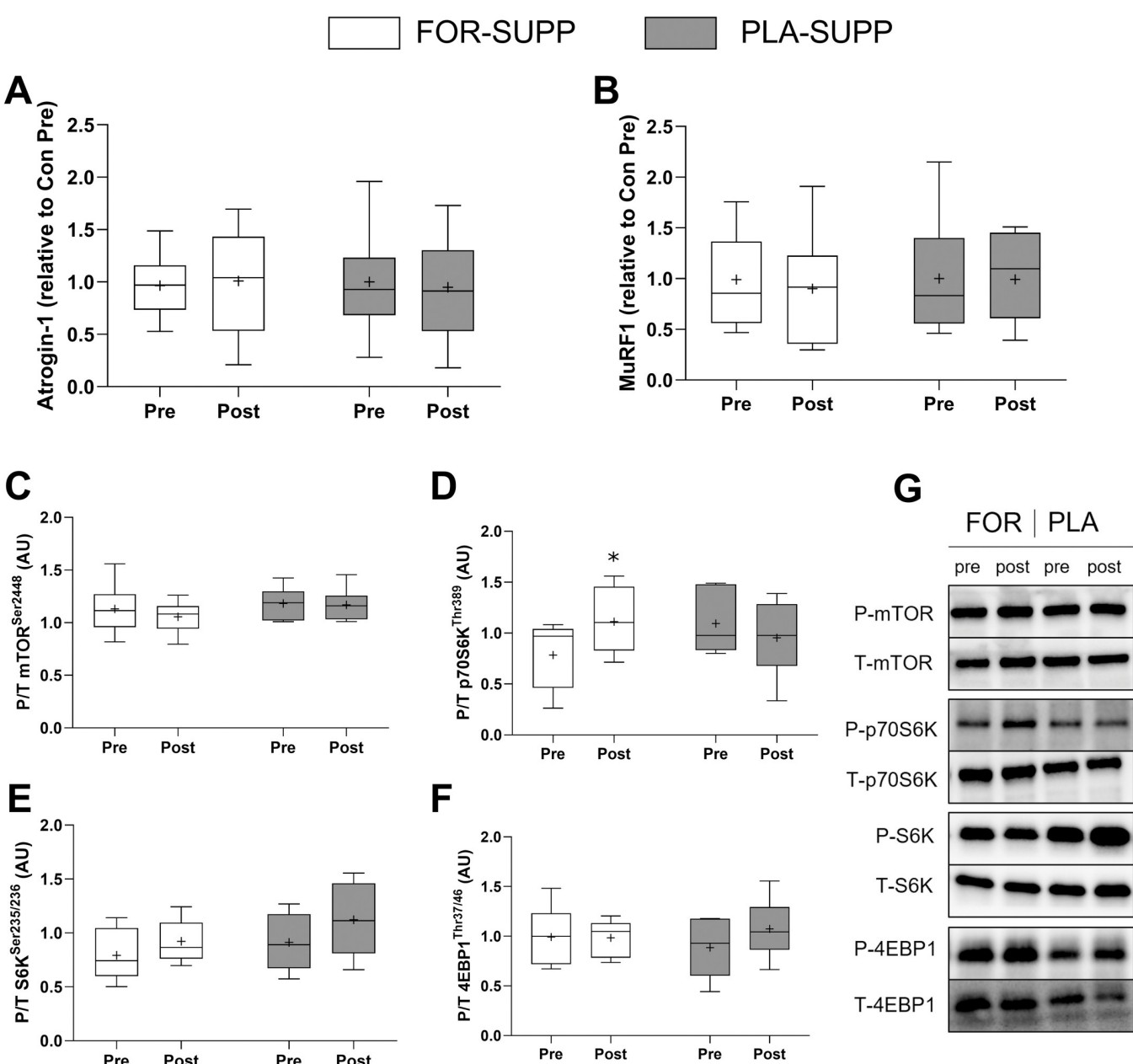

**Fig 7.** Atrogin-1 (A) and MuRF1 (B) gene expression, and the ratio of phosphorylated to total protein of mTOR (C), p70S6K (D), S6K (E), and 4EPB1 at pre-(day 14) and post- (day 28) immobilization. Representative western blot bands (G). Boxes represent the $25^{th}$ to $75^{th}$ percentile, with the central horizontal lines indicating the median. The whiskers represent the maximal and minimal values, and the cross represents the mean value. 9 and 6 participants in FOR-SUPP and PLA-SUPP, respectively, were analyzed for protein expression due to shortage of muscle tissues. FOR-SUPP, Fortetropin® supplement; PLA-SUPP, placebo supplement; CSA, cross-sectional area. * $P < 0.05$, significantly different from pre within the same group.

normal physical activity. In line with the results of CSA and LM of the immobilized leg, the molecular mechanisms we probed (the number of myostatin-positive SC and the expression of E3 ubiquitin ligases) showed no effect of FOR supplementation, except that phosphorylation of p70S6K was increased post-immobilization in the FOR-SUPP. Isometric peak torque decreased following two weeks of single-leg immobilization; however, two weeks of returning to normal activity was sufficient to recover the reduction in peak torque.

Myostatin is a potent negative regulator of muscle growth, and an increased circulating concentration of myostatin has been observed in various conditions, including disease or disuse states [28–31]. Previous studies showed unchanged myostatin mRNA [12] or decreased myostatin protein expression [32] in single-leg immobilization model. However, we observed increased plasma myostatin in the PLA-SUPP after the 6-wk protocol, including 2 weeks of single-leg immobilization (Fig 5). On the other hand, the FOR-SUPP did not show a rise (Fig 5). These results are similar to previous studies that reported a positive effect of FOR ingestion on blood myostatin levels in humans [13] and dogs [15]. Furthermore, the percentage change of myostatin concentration in plasma or the circulating myostatin level at day 42 showed a moderate association with the percentage change in CSA measured by ultrasonography, regardless of group (Fig 5C and 5D). Evans et al. [14] reported a positive association between circulating myostatin concentration and MPS on Day 21 of treatments (FOR vs. placebo consumption). These authors [14] suggested that the elevated MPS may result from the increased amino acid availability derived from protein degradation. In the current study, despite preventing myostatin levels from rising, FOR ingestion did not attenuate the loss of muscle size and LM of the immobilized leg during the immobilization phase (Fig 2).

Myostatin circulates in two forms, latent and active, and only activated C-terminal myostatin, after proteolytic processing, can bind to the ActRIIB, thereby initiating signaling, including the pathways activating UPS and inhibiting myogenesis [8]. Thus, one possibility to explain the discordance between myostatin circulation level and muscle mass in the present study is that the ELISA evaluating changes in myostatin concentration might not be sensitive enough to distinguish the different forms of myostatin, as Bergen et al. proposed [33]. In the present study, the molecular kinetics of myostatin activity between blood and muscle at the cellular level are unclear. However, in line with the results of muscle size and LM of the immobilized leg, the expression of MuRF1 and Atrogin-1 (Fig 7A and 7B) were not different between conditions, and neither were the number of myostatin-positive SC (Fig 6D and 6E) between groups at any time points. The number of myostatin-positive SC–myostatin inhibits the progress of SC into the S phase of the cell cycle [9]–increased after 2 weeks of immobilization in both groups. More studies are needed to determine the relationship between disuse-atrophy and myostatin-positive SC. However, it was in line with the previous study reporting the blunted response of the decline in the number of myostatin-positive SC to resistance exercise in older adults, who may have an impaired myogenic capacity compared to young [34]. These results show the role of myostatin in the regulation of SC function. Besides that, we observed the increased phosphorylated p70S6K at post-immobilization in FOR-SUPP (Fig 7D), which should not be disregarded.

The mechanisms of FOR on skeletal muscle growth have not been completely determined. However, egg yolk contains macro- and micro-nutrients (lipid, minerals, vitamins, micro-RNAs, and more) that may modify pathways related to MPS or degradation [35, 36]. Indeed, Sharp et al. [13] reported that 12 weeks of FOR supplement increased LM and decreased protein degradation markers, increasing mTOR signaling activity in trained young men. Also, Evans et al. [14] showed elevated MPS in older adults who consumed FOR for 21 days, despite no increase in LM. Contrary to our study, these previous studies did not restrict participants' mobility or physical activity level during FOR consumption [13, 14]. The influence of FOR on muscle metabolism may have been relatively subtle in our study due to the absence of muscular contraction.

After disuse atrophy was induced, the decreased muscle CSA and LM of the immobilized leg were not recovered within 2 weeks of returning to normal activity in both groups (Fig 2). Similar to the period of immobilization, we could not observe any impact of FOR ingestion on skeletal muscle during the recovery phase. A previous study reported that 2 weeks of

immobilization was sufficient to induce anabolic resistance–decreased response of MPS to anabolic stimuli–in young adults [16]. Thus, there is a possibility that the impact of FOR was not effective enough to recover the loss of muscle size with anabolic resistance induced by immobilization or needed an extended recovery period. Suetta et al. [37] reported that 4 weeks of resistance exercise rehabilitation recovered 2-wk of immobilization-induced loss of muscle mass in young men; however, given the volume of work and loads lifted during the recovery period, these findings [37] are perhaps unsurprising. In the present study, our participants did not perform any exercise-type rehabilitation during the recovery phase, although they returned to their normal physical activity levels. These results show the necessity of exercise-based rehabilitation to recover disuse-induced loss of muscle mass in the short term.

Contrary to muscle mass, the decreased leg extension peak torque was largely recovered by 2 weeks of normal activity recovery (Fig 4). Generally, reduced muscle mass induced by immobilization is accompanied by a decline in muscle strength and impairment of muscle function [38, 39]. However, the amount of muscle mass does not always align with muscle strength, and neural adaptations are likely involved in determining muscle strength [40]. Sarto et al. [41] reported impaired neuromuscular junction (NMJ) stability, increased motor unit potential complexity, and decreased motor unit firing rate after 10 days of single leg immobilization, but the impaired neural adaptations were fully recovered by 21 days of recovery accompanied resistance exercise. The authors [41] suggested a functional resilience of human NMJ against disuse-induced stress. Thus, strength could be more flexible and recover faster than muscle mass [42]. In line with this thesis, we noted that the resumption of 2 weeks of normal activity recovery was sufficient to reverse the, presumably, neural-level adaption impaired by immobilization, thereby recovering muscle strength.

In the present study, daily ingestion of 19.8 g FOR (6wk) did not adversely affect blood variables (Table 3). However, a significant decline in ASP occurred in both groups; however, the values on both days 1 and 42 were still well within the normal range—5 to 30 U/L [43]. In line with our results, a previous study [13] reported no changes in the blood variables during 12 weeks of FOR ingestion.

Due to the COVID-19 outbreak and the poor compliance with wearing a knee brace, we could not analyze the total number of participants planned to have a medium effect size of power (S1 File) and recognize that our findings are limited to young men. We did not collect blood samples on days 14 and 28.

## Conclusions

We showed that FOR ingestion was safe and tolerable, and it prevented the rise in myostatin observed after the 6-wk protocol, which included 2 weeks of single-leg immobilization. However, the unchanged myostatin circulation concentration in the FOR group did not affect the molecular markers related to myostatin-regulated signaling in skeletal muscle and did not attenuate disuse-induced muscle atrophy. We note that 2 weeks of normal activity following two weeks of immobilization was insufficient to recover the decline in muscle CSA and peak torque during disuse.

## Supporting information

**S1 Fig. A representative image of vastus lateralis CSA obtained by ultrasonography.** VL, vastus lateralis; VI, vastus intermedius, RF, rectus femoris.
(TIF)

**S2 Fig. Representative images of skeletal muscle fiber type.** (A) merge of all the panels; (B) Type I fibers, purple; (C) Type II fibers, green; (D) Laminin, blue.
(TIF)

**S1 File. Consolidated standards of Reporting Trials flow diagram.** FOR-SUPP, Fortetropin® supplement; PLA-SUPP, placebo supplement.
(DOC)

**S2 File. Consolidated standards of reporting trials chart.**
(DOC)

**S3 File. Minimal dataset.**
(XLSX)

**S4 File. Western blot raw images.**
(PDF)

## Acknowledgments

We want to thank Prior Todd for technical and administrative support and the participants for their time and effort.

## Author Contributions

**Conceptualization:** Changhyun Lim, Stuart M. Phillips.

**Data curation:** Changhyun Lim, James McKendry, Taylor Giacomin, Jonathan C. Mcleod, Brad S. Currier, Giulia Coletta, Stuart M. Phillips.

**Formal analysis:** Changhyun Lim, Taylor Giacomin, Sean Y. Ng, Stuart M. Phillips.

**Funding acquisition:** Stuart M. Phillips.

**Investigation:** Changhyun Lim, James McKendry, Stuart M. Phillips.

**Methodology:** Changhyun Lim, Sean Y. Ng, Stuart M. Phillips.

**Supervision:** Stuart M. Phillips.

**Writing – original draft:** Changhyun Lim.

**Writing – review & editing:** Changhyun Lim, James McKendry, Taylor Giacomin, Jonathan C. Mcleod, Sean Y. Ng, Brad S. Currier, Giulia Coletta, Stuart M. Phillips.

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
