## [Decision Letter · Decision Letter 0]

22 Feb 2023

PONE-D-22-33716Fortetropin® supplementation prevents the rise in circulating myostatin but not disuse-induced muscle atrophy in young men during immobilization: a randomized controlled trialPLOS ONE

Dear Dr. Phillips,

Thank you for submitting your manuscript to PLOS ONE. After careful consideration, we feel that it has merit but does not fully meet PLOS ONE’s publication criteria as it currently stands. Therefore, we invite you to submit a revised version of the manuscript that addresses the points raised during the review process.

We look forward to receiving your revised manuscript.

Kind regards,

Krzysztof Durkalec-Michalski, Ph.D

Academic Editor

PLOS ONE

Journal Requirements:

“I have read the journal's policy and the authors of this manuscript have the following competing interests: SMP reports grants or research contracts from the US National Dairy Council, Canadian Institutes for Health Research, Dairy Farmers of Canada, Roquette Freres, Ontario Centre of Innovation, Nestle Health Sciences, Myos, National Science and Engineering Research Council and the US NIH during the conduct of the study; personal fees from Nestle Health Sciences, non-financial support from Enhanced Recovery, outside the submitted work. SMP has patents licensed to Exerkine but reports no financial gains from any patent or related work. Other authors have declared that no competing interests exist.”

Reviewers' comments:

Reviewer's Responses to Questions

**Comments to the Author**

1. Is the manuscript technically sound, and do the data support the conclusions?

Reviewer #1: Partly

Reviewer #2: Partly

2. Has the statistical analysis been performed appropriately and rigorously? 

Reviewer #1: Yes

Reviewer #2: No

3. Have the authors made all data underlying the findings in their manuscript fully available?

Reviewer #1: Yes

Reviewer #2: Yes

4. Is the manuscript presented in an intelligible fashion and written in standard English?

Reviewer #1: Yes

Reviewer #2: No

5. Review Comments to the Author

Reviewer #1: This study aimed to investigate the impact of Fortetropin® (FOR) supplementation on immobilization-induced muscle atrophy and weakness. The research question is relevant due to the increasing interest in the scientific literature on disuse countermeasures and the promising findings of FOR supplementation in previous training studies. The authors employed an appropriate disuse model (2-week knee brace immobilization) with subsequent recovery, controlled for/assessed potential confounding factors (diet and physical activity levels) and evaluated alterations in molecular markers involved in muscle atrophy. Overall, the manuscript is clear and easy to follow. However, there are some crucial matters to consider regarding plasma myostatin concentration evaluation.

First, blood samples were collected only on day 1 (onset of the run-in phase) and day 42 (end of the recovery). Due to this limitation in study design, the authors cannot determine with certainty whether the increased myostatin concentration occurred during the immobilization phase or the recovery phase, or both. This is particularly relevant in light of a recent 10-day bed rest study that found an unexpected increase in circulating myostatin after two days of recovery, but not during the unloading period (Oranger et al., 2022). Why did the authors decide to not collect blood samples on day 14 (onset of the immobilization phase) and day 28 (end of the immobilization phase)? Blood sampling is a relatively fast procedure and less invasive compared to muscle biopsies that were obtained also at these two additional time points.

Second, caution is needed in the interpretation of findings exclusively drawn from circulating biomarkers when employing “local” disuse models, such as knee brace immobilization. Indeed, disuse effects are expected mostly at a local muscle level and results obtained from systemic biomarkers may be affected by the relatively low muscle mass undergoing unloading/immobilization (one leg only).

To overcome these issues, have the authors considered evaluating myostatin transcript and protein levels from muscle biopsies? I strongly recommend performing these additional analyses, if possible.

Other comments:

L1: In the title, the authors suggest that the rise in myostatin occurred during the immobilization phase. However, since blood samples were not collected on days 14 and 28 the authors cannot be completely sure of this (see my main comment).

L28: If word count allows it, I suggest the authors specify that the recovery phase in this study consisted just of the resumption of habitual physical activities.

L34-39: The authors reported just one percentage of reduction and p-value for each muscle force and mass parameter. To which group are they referred (FOR-SUPP or PLA-SUPP)? Or is it an average?

L39-40: As for the title, I believe the authors cannot affirm that the rise in myostatin is due to the immobilization period.

L59: I feel some background on the role of myostatin in disuse atrophy is needed here in the introduction. Is increased myostatin an established finding with muscle disuse/immobilization? To the best of my knowledge, there is conflicting evidence on the effect of human limb disuse/immobilization on myostatin levels. Previous studies employing cast immobilization found unchanged (Jones et al., 2004; Chen et al., 2007) or increased (Dirks et al., 2014) MSNT expression, or even decreased myostatin protein content (Snijders et al., 2014). Increased myostatin gene and protein levels were observed in response to 3 days of unilateral lower limb suspension (ULLS) (Gustafsson et al., 2010), but no changes were found in a longer 14-day ULLS (de Boer et al., 2007). Finally, 25 days (Zachwieja et al., 1999), but not 10 days (Oranger et al., 2022) of bed rest increased circulating myostatin levels. Results in animal disuse studies seem also ambiguous (Carlson et al., 1999; Wehling et al., 2000; Allen et al., 2009). I believe the authors should cite and discuss at least part of this previous literature in the manuscript to give to the reader a broader background.

L82: Authors should add DXA procedures to the Methods section or at least refer the reader to previous studies using the same procedures.

L145: How was the amount of daily FOR supplementation (19.8g/d) determined?

L168: Were the subjects asked to refrain from intense physical activity during the whole protocol (run-in + immobilization + recovery) or just during the immobilization phase?

L180-181: Are the ultrasound-derived CSA values presented in the article the average of these 7-9 images? Why did you choose to collect the images at 2cm intervals, instead of capturing different muscle regions (e.g. 70%, 60%, 50%, 40% 30%, 20% of femur length) to take into account potential regional differences?

L210: Could you please provide more information regarding muscle biopsies exact location and precautions to avoid the effects of pre-sampling?

L313: How was participants’ compliance evaluated?

L332: Have the authors collected any functional or morphological data on the control leg? I think it would be a nice addition to the manuscript.

L364: I suggest the authors state here that no differences were observed in absolute CSA values of Type I fibres, as reported in Figure 3A.

L454: The authors are invited to report also in the main text that WB was performed only on a subgroup of participants (N=9 for FOR-SUPP and n=6 for PLA-SUPP) due to shortage of muscle tissue.

L455: Add “(data not shown)” after “E3 ubiquitin ligases”, as this result is not reported in any figures or tables.

L480: Did E3 ubiquitin ligases change their expression with immobilization? In the results section (L455), the authors reported only that no interactions between group and time for E3 ubiquitin ligases were found, but here in the discussion, the authors suggest that is one of the proved molecular mechanisms involved in muscle atrophy with disuse.

L486: I suggest the authors discuss here that the literature however is not always in agreement and different studies found unchanged or decreased myostatin levels in disuse scenarios (Jones et al., 2004; de Boer et al., 2007; Snijders et al., 2014; Oranger et al., 2022).

L507: While MuRF1 and Atrogin-1 expressions did not differ between conditions, they were also unaltered by immobilization, thus I am not sure these results are in line with the reduced muscle size and LM observed.

L510-511: Did the authors expect this increase in myostatin-positive SC with immobilization? I believe this novel finding should be further discussed.

L543: The authors forgot to add “complexity” after “motor unit potential”. In addition, I suggest specifying that the retraining period of the cited study was based on resistance training, differently from the present manuscript.

L614: please fix reference number 12, probably an error with the reference manager occurred.

References:

- Allen DL, Bandstra ER, Harrison BC, Thorng S, Stodieck LS, Kostenuik PJ, Morony S, Lacey DL, Hammond TG, Leinwand LL, Argraves WS, Bateman TA & Barth JL (2009). Effects of spaceflight on murine skeletal muscle gene expression. J Appl Physiol 106, 582–592.

- de Boer MD, Selby A, Atherton P, Smith K, Seynnes OR, Maganaris CN, Maffulli N, Movin T, Narici M V. & Rennie MJ (2007). The temporal responses of protein synthesis, gene expression and cell signalling in human quadriceps muscle and patellar tendon to disuse. J Physiol 585, 241–251.

- Carlson CJ, Booth FW & Gordon SE (1999). Skeletal muscle myostatin mRNA expression is fiber-type specific and increases during hindlimb unloading. Am J Physiol - Regul Integr Comp Physiol 277, 601–606.

- Chen YW, Gregory CM, Scarborough MT, Shi R, Walter GA & Vandenborne K (2007). Transcriptional pathways associated with skeletal muscle disuse atrophy in humans. Physiol Genomics 31, 510–520.

- Dirks ML, Wall BT, Snijders T, Ottenbros CLP, Verdijk LB & Van Loon LJC (2014). Neuromuscular electrical stimulation prevents muscle disuse atrophy during leg immobilization in humans. Acta Physiol 210, 628–641.

- Gustafsson T, Osterlund T, Flanagan JN, Von Waldén F, Trappe TA, Linnehan RM & Tesch PA (2010). Effects of 3 days unloading on molecular regulators of muscle size in humans. J Appl Physiol 109, 721–727.

- Jones SW, Hill RJ, Krasney PA, O’Conner B, Peirce N & Greenhaff PL (2004). Regulation of skeletal muscle mass and adipose tissue mass by follistatin and follistatin-related gene (FLRG) and development of novel polypeptides as medical drugs. FASEB J; DOI: 10.1096/fj.03-1228fje.

- Oranger A, Storlino G, Dicarlo M, Zerlotin R, Pignataro P, Sanesi L, Narici M, Pišot R, Simunič B, Colaianni G, Grano M & Colucci S (2022). Impact of 10-day bed rest on serum levels of irisin and markers of musculoskeletal metabolism. FASEB J 37, e22668.

- Snijders T, Wall BT, Dirks ML, Senden JMG, Hartgens F, Dolmans J, Losen M, Verdijk LB & Van Loon LJC (2014). Muscle disuse atrophy is not accompanied by changes in skeletal muscle satellite cell content. Clin Sci 126, 557–566.

- Wehling M, Cai B & TidballL J (2000). Modulation of myostatin expression during modified muscle use. FASEB J 14, 103–110.

- Zachwieja JJ, Smith SR, Sinha-Hikim I, Gonzalez-Cadavid N & Bhasin S (1999). Plasma myostatin-immunoreactive protein is increased after prolonged bed rest with low-dose T3 administration. J Gravitational Physiol.

Reviewer #2: Major revisions

1. Sample size determination:

a) Line 104: please provide the references for the effect size interpretation;

b) Line 105: it is indicated that total calculated sample size is 24 (12 participants per each group). While, the final statistical analysis covered 11 participants is FOR-SUPP group and 9 participants in PLA-SUPP groups. In this circumferences, what were the actual effect size, alpha and power? Was the final sample size sufficient? Why Authors did not accounted for possible drop outs at the stage of a priori sample size determination? Western blot analysis was performed for even lower number of participants.

2. What was the reason for performing blood evaluations ‘solely’ at days 1 and 42? Why were they not performed at days 14 and 28?

3. Lines 143-145: how was the nutritional value of Fortetropin supplement evaluated? Please provide methodology and/or relevant references.

4. Line 161: With respect to protein intake during the immobilization phase (1.2 g protein per kg of body mass), Authors are referring to the paper published in 2022 (doi: 10.1002/jcsm.12922), while the study on Fortetropin supplementation started in 2020. Please provide the references/guidelines that actually were implemented when establishing protein intake during immobilization phase.

5. It was mentioned that energy requirements during immobilization phase were calculated by multiplying basal metabolic rate (as calculated with Harris-Benedict equation) by physical activity level. Further Authors indicated that energy expenditures were monitored using Actigraph wGT2X-BT activity monitor. Thus, is it possible to compare the calculated/pre-assumed energy expenditures and the actual/established with Actigraph energy expenditures? What was the actual energy balance in participants during immobilization phase?

6. Lines 337-352: It is not clear to which group (FOR-SUPP or PLA-SUPP) the provided results (CSA, LM, and changes in these indices) refer to? It must be clarified and specified. Are the values means or medians?

7. Lines 362-374: It is not clear to which group (FOR-SUPP or PLA-SUPP) the provided results (muscle fiber CSA) refer to? It must be clarified and specified. Although, there were no significant differences between groups, still it is impossible that the results are exactly the same in both groups. Maybe be it would be reasonable to decline from providing the exact mean results/values in the text (leave ‘P’ values only) – while the exact results are provided in the corresponding figures. The figures are very clear and highly-informative.

8. Please refer point 7 to results on isometric knee-extensor torque.

Minor revisions

1. Initial concentrations of plasma myostatin were 5487 ± 489 pg/mL in FOR-SUPP group and 4221 ± 541 pg/mL in PLA-SUPP group. Thus, was there a significant difference in baseline plasma myostation between groups? Please correct decimal places in lines 409-410 (results section)

2. Please standardize the way of recording of ‘Fortetropin’ name throughout the manuscript – standardize to use or not to use ‘®’ sign.

3. Line 40: the following sing ‘-‘ is lacking the writing of ‘2-wk’.

4. Please standardize using or not to using spaces between special signs such as ‘=/±/>/<!--≤/≥’ throughout the manuscript. Correct the using/not using the spaces between values and units throughout the manuscript.<br /5. Please standardize the way of recording of ‘vs.’ or ‘vs’ throughout the manuscript.

6. Line 108-111: In general, this part of the text is unclear and needs to be reworded.

7. Line 119, 471: Consider not using first person expressions.

8. Line 126: Precise the writing of the hours of arriving to the laboratory.

9. Line 166: Correct the writing of word ‘exercise’.

10. Standardize the writing of the names of study phases throughout the manuscript – sometimes capitals are used as a first letter of the name of the phases, and in the other parts of the manuscript lowercase is used.

11. Lines 168-169: Correct and standardize the writing of units (i.e., ‘steps/d-1; energy expenditure, kcal/d; metabolic equivalents of task [METs/d-1]’). It also refers to Table 2.

12. Line 194: Maybe it is better to write ‘3 trials’ if you are using the numbers in the earlier part of the paragraph (i.e., 4 repetitions).

13. Lines 204-205: What is the rational to use capitals for ‘Creatinine’, ‘Glucose’ etc.?

14. Line 306-307: Part of results are expressed as median and percentile ranges.

15. The references list it messy. It must be corrected according to PLoS One guidelines.

16. Please consider emergence ‘Conclusions’ from ‘Discussion section’.

6. PLOS authors have the option to publish the peer review history of their article (what does this mean?). If published, this will include your full peer review and any attached files.

Reviewer #1: **Yes: **Fabio Sarto

Reviewer #2: No

---

## [Author Response · Author response to Decision Letter 0]

14 Mar 2023

We have uploaded it as a separate file labeled 'Response to Reviewers'.

---

## [Decision Letter · Decision Letter 1]

26 Apr 2023

PONE-D-22-33716R1Fortetropin® supplementation prevents the rise in circulating myostatin but not disuse-induced muscle atrophy in young men with limb immobilization: a randomized controlled trialPLOS ONE

Dear Dr. Phillips,

Thank you for submitting your manuscript to PLOS ONE.After careful consideration, we feel that it has merit but does not fully meet PLOS ONE’s publication criteria as it currently stands. Therefore, we invite you to submit a revised version of the manuscript that addresses the points raised during the review process.

We look forward to receiving your revised manuscript.

Kind regards,

Krzysztof Durkalec-Michalski, Ph.D

Academic Editor

PLOS ONE

Journal Requirements:

Reviewers' comments:

Reviewer's Responses to Questions

**Comments to the Author**

1. If the authors have adequately addressed your comments raised in a previous round of review and you feel that this manuscript is now acceptable for publication, you may indicate that here to bypass the “Comments to the Author” section, enter your conflict of interest statement in the “Confidential to Editor” section, and submit your "Accept" recommendation.

Reviewer #1: (No Response)

Reviewer #2: (No Response)

2. Is the manuscript technically sound, and do the data support the conclusions?

Reviewer #1: Yes

Reviewer #2: Yes

3. Has the statistical analysis been performed appropriately and rigorously? 

Reviewer #1: Yes

Reviewer #2: Yes

4. Have the authors made all data underlying the findings in their manuscript fully available?

Reviewer #1: Yes

Reviewer #2: Yes

5. Is the manuscript presented in an intelligible fashion and written in standard English?

Reviewer #1: Yes

Reviewer #2: Yes

6. Review Comments to the Author

Reviewer #1: The authors addressed most of my concerns.

In my opinion, the collection of blood samples only on days 14 and 28 should be clearly stated as a limitation of the manuscript.

Other minors comments:

L62: it's "Jones et al"

L537: add please the reference

Reviewer #2: Minor revisions

1. The reference list is still in mess. It must be improved.

2. The writing of values and units still require correction within the whole manuscript. Apart from ‘%’, Authors should use put space between the value and the unit. Unify the way of writing of units within the whole manuscript and in the figures/tables.

3. Line 62: Put ‘.’ after ‘et al.’.

4. What is the rational to use capitals for ‘Creatinine’, ‘Glucose’ etc.? – it should be corrected within the whole manuscript.

5. Lines 140-141: ‘All data were stored in a locked filing cabinet and a computer protected 141 by a password’ the sentence seems to be not necessary. It can be removed.

6. Line 163: Replace ‘consumed’ with ‘intake’.

7. Line 225: Replace ‘tissues’ with ‘samples’.

8. Replace ‘minutes’ with ‘min’ within the whole manuscript.

9. Lines 300 – 301: The space between these lines is wider compared to other lines.

10. Lines 333 and Table 2: I suggest replacing ‘activity’ with ‘energy expenditure related to/arising from physical activity’ or with similar expression. The expression ‘activity’ is misleading.

11. Line 355: ‘The value of immobilized leg LM showed a similar trend as vastus lateralis’ the sentence needs to be reformulated and improved. In the current version it is not clear.

12. Line 387: Pleas put the word ‘between’ before ‘groups’.

13. Regarding the remark from the first round of the review:

“Lines 337-352: It is not clear to which group (FOR-SUPP or PLA-SUPP) the provided results (CSA, LM, and changes in these indices) refer to? It must be clarified and specified. Are the values means or medians? Thank you. Because there was no time-by-group interaction, we described the data of the main effect for time using the values combined in both groups, with the saying ‘no difference between groups.’ We have revised the sentences to make them clearer for readers on lines 350-362. All data are presented as mean +/- SD.”

Please provide the information that you used mean value from both groups at the beginning of each paragraph within which the procedure was implemented.

14. Please verify if all the units provided in the Table 3 are correct.

15. Line 505: Remove ‘There were” from the beginning of the sentence.

16. Lines 541, 544: Authors can use ‘MPS’ instead of ‘muscle protein synthesis’.

7. PLOS authors have the option to publish the peer review history of their article (what does this mean?). If published, this will include your full peer review and any attached files.

Reviewer #1: **Yes: **Fabio Sarto

Reviewer #2: No

---

## [Author Response · Author response to Decision Letter 1]

28 Apr 2023

Thank you for the reviews. We have made revisions to our paper and offer the following respnoses to the reviewers’ points.

Reviewer #1: The authors addressed most of my concerns.

In my opinion, the collection of blood samples only on days 14 and 28 should be clearly stated as a limitation of the manuscript.

We have clearly stated that we did not collect blood samples on days 14 and 28 on lines 587-588. 

Other minors comments:

L62: it's "Jones et al"

Thank you. We have corrected ‘Jone’ to “Jones” on line 62. 

L537: add please the reference

We have cited the reference on line 540. 

Reviewer #2: Minor revisions

1. The reference list is still in mess. It must be improved.

All references are inserted using EndNote software and all are corrected at the typsetting stage of manuscript preparation (by the publisher). Nonetheless, we have corrected the format of reference list according to the PLOS ONE guidelines. 

2. The writing of values and units still require correction within the whole manuscript. Apart from ‘%’, Authors should use put space between the value and the unit. Unify the way of writing of units within the whole manuscript and in the figures/tables.

We have corrected this throughout the manuscript and in the figures 2, 3, 4, 5, and 6. 

3. Line 62: Put ‘.’ after ‘et al.’.

We have put ‘.’ after et al on line 62. 

4. What is the rational to use capitals for ‘Creatinine’, ‘Glucose’ etc.? – it should be corrected within the whole manuscript.

Thanks for highlighting this. We have corrected the capitals to lower case throughout the manuscript. 

5. Lines 140-141: ‘All data were stored in a locked filing cabinet and a computer protected 141 by a password’ the sentence seems to be not necessary. It can be removed.

We have removed the sentence on line 140. 

6. Line 163: Replace ‘consumed’ with ‘intake’.

We have replaced ‘consumed’ with ‘intake’ on line 163.

7. Line 225: Replace ‘tissues’ with ‘samples’.

We have replaced ‘tissues’ with ‘samples’ on line 226. 

8. Replace ‘minutes’ with ‘min’ within the whole manuscript.

We have replaced ‘minutes’ with ‘min’ throughout the manuscript. 

9. Lines 300 – 301: The space between these lines is wider compared to other lines.

We have corrected the space to be same as others by changing the symbol, � on line 302. 

10. Lines 333 and Table 2: I suggest replacing ‘activity’ with ‘energy expenditure related to/arising from physical activity’ or with similar expression. The expression ‘activity’ is misleading.

We have changed ‘activity’ with ‘energy expenditure by physical activity’ in Table 2 and on lines 335-340.

11. Line 355: ‘The value of immobilized leg LM showed a similar trend as vastus lateralis’ the sentence needs to be reformulated and improved. In the current version it is not clear.

We have removed the sentence to avoid any confusion caused by it on line 359. 

12. Line 387: Pleas put the word ‘between’ before ‘groups’.

We have added ‘between’ before ‘groups’ on line 391. 

13. Regarding the remark from the first round of the review:

“Lines 337-352: It is not clear to which group (FOR-SUPP or PLA-SUPP) the provided results (CSA, LM, and changes in these indices) refer to? It must be clarified and specified. Are the values means or medians? Thank you. Because there was no time-by-group interaction, we described the data of the main effect for time using the values combined in both groups, with the saying ‘no difference between groups.’ We have revised the sentences to make them clearer for readers on lines 350-362. All data are presented as mean +/- SD.”

Please provide the information that you used mean value from both groups at the beginning of each paragraph within which the procedure was implemented.

We have clearly stated that it’s the mean value of the absolute CSA and immobilization leg LM on lines 353 and 363.

14. Please verify if all the units provided in the Table 3 are correct.

Thanks for highlighting this. We have corrected IU/L to U/L throughout the manuscript. 

15. Line 505: Remove ‘There were” from the beginning of the sentence.

Thanks for highlighting this. We have removed ‘There were’ on line 509. 

16. Lines 541, 544: Authors can use ‘MPS’ instead of ‘muscle protein synthesis’.

We have revised ‘muscle protein synthesis’ to ‘MPS’ on line 545 and 548.

---

## [Decision Letter · Decision Letter 2]

11 May 2023

Fortetropin® supplementation prevents the rise in circulating myostatin but not disuse-induced muscle atrophy in young men with limb immobilization: a randomized controlled trial

PONE-D-22-33716R2

Dear Prof. Philips,

We’re pleased to inform you that your manuscript has been judged scientifically suitable for publication and will be formally accepted for publication once it meets all outstanding technical requirements.

Kind regards,

Krzysztof Durkalec-Michalski, Ph.D

Academic Editor

PLOS ONE

Additional Editor Comments (optional):

Reviewers' comments:

Reviewer's Responses to Questions

**Comments to the Author**

1. If the authors have adequately addressed your comments raised in a previous round of review and you feel that this manuscript is now acceptable for publication, you may indicate that here to bypass the “Comments to the Author” section, enter your conflict of interest statement in the “Confidential to Editor” section, and submit your "Accept" recommendation.

Reviewer #1: All comments have been addressed

Reviewer #2: All comments have been addressed

2. Is the manuscript technically sound, and do the data support the conclusions?

Reviewer #1: Yes

Reviewer #2: Yes

3. Has the statistical analysis been performed appropriately and rigorously? 

Reviewer #1: Yes

Reviewer #2: Yes

4. Have the authors made all data underlying the findings in their manuscript fully available?

Reviewer #1: Yes

Reviewer #2: Yes

5. Is the manuscript presented in an intelligible fashion and written in standard English?

Reviewer #1: Yes

Reviewer #2: Yes

6. Review Comments to the Author

Reviewer #1: The authors addressed my all of my concerns and added the limitation requested.

I have no further comments.

Reviewer #2: (No Response)

7. PLOS authors have the option to publish the peer review history of their article (what does this mean?). If published, this will include your full peer review and any attached files.

Reviewer #1: **Yes: **Fabio Sarto

Reviewer #2: No

---

## [Editor Report · Acceptance letter]

15 May 2023

PONE-D-22-33716R2 

Fortetropin supplementation prevents the rise in circulating myostatin but not disuse-induced muscle atrophy in young men with limb immobilization: a randomized controlled trial 

Dear Dr. Phillips:

I'm pleased to inform you that your manuscript has been deemed suitable for publication in PLOS ONE. Congratulations! Your manuscript is now with our production department. 

Kind regards, 

on behalf of

Dr. Krzysztof Durkalec-Michalski 

Academic Editor

PLOS ONE